# AlphaFold2 and RoseTTAFold predict posttranslational modifications. Chromophore formation in GFP-like proteins

**Sophia M. Hartley[1‡], Kelly A. Tiernan[1‡], Gjina Ahmetaj[1‡], Adriana Cretu[1‡], Yan Zhuang[2], Marc Zimmer[1]***

**1** Department of Chemistry, Connecticut College, New London, CT, United States of America, **2** Department of Mathematics and Statistics, Connecticut College, New London, CT, United States of America

‡ SMH and KAT authors contributed equally, first authors, names in alphabetic order. GA and AC authors contributed equally, names in alphabetic order.
* mzim@conncoll.edu

**Data Availability Statement:** All relevant data are within the paper and its Supporting Information files.

## Abstract

AlphaFold2 and RoseTTAfold are able to predict, based solely on their sequence whether GFP-like proteins will post-translationally form a chromophore (the part of the protein responsible for fluorescence) or not. Their training has not only taught them protein structure and folding, but also chemistry. The structures of 21 sequences of GFP-like fluorescent proteins that will post-translationally form a chromophore and of 23 GFP-like non-fluorescent proteins that do not have the residues required to form a chromophore were determined by AlphaFold2 and RoseTTAfold. The resultant structures were mined for a series of geometric measurements that are crucial to chromophore formation. Statistical analysis of these measurements showed that both programs conclusively distinguished between chromophore forming and non-chromophore forming proteins. A clear distinction between sequences capable of forming a chromophore and those that do not have the residues required for chromophore formation can be obtained by examining a single measurement—the RMSD of the overlap of the central alpha helices of the crystal structure of S65T GFP and the AlphaFold2 determined structure. Only 10 of the 578 GFP-like proteins in the pdb have no chromophore, yet when AlphaFold2 and RoseTTAFold are presented with the sequences of 44 GFP-like proteins that are not in the pdb they fold the proteins in such a way that one can unequivocally distinguish between those that can and cannot form a chromophore.

## Introduction

AlphaFold2 [1, 2] and RoseTTAfold [3] are two freely available programs that can predict three-dimensional protein structures from their amino acid sequence with atomic accuracy. Both programs were created by machine learning and the ~180,000 structures in the protein data bank (pdb) [4, 5] were used as an important training set. The three-dimensional structures of many of the protein structures in the pdb have been influenced by the proteins binding ligands and post-translation modifications. Agirre and co-workers have shown that

**Funding:** The author(s) received no specific funding for this work.

**Competing interests:** The authors have declared that no competing interests exist.

AlphaFold2 predicts the folding of glycosylated proteins solely from the amino acid sequence of the proteins because the program was trained on the pdb in which the glycosylated proteins are found in their full or partially glycanated forms [6]. In the absence of its heme cofactor AlphaFold2 will fold a hemoglobin subunit so that it has a cavity complementary with a heme subunit [6, 7] and it will identify iron-sulfur cluster and zinc binding sites [8]. Finally, Alpha-Fold2 has been shown to generate protein-peptide complex structures without multiple alignments of the peptide fragment [9].

Fluorescent Proteins (FPs) are commonly used molecular tracer molecules. A Nobel Prize was awarded in 2008 for the development of GFP as a "tagging tool in bioscience" [10] and, in 2014, for the use of fluorescent proteins in "the development of super-resolved fluorescence microscopy" [11]. Four books [12–15] and numerous reviews [16–21] have been written about fluorescent proteins. It is their post-translational autocatalytic chromophore formation that makes these genetic tracer molecules so useful. They are all folded into a β-barrel composed of eleven β sheets surrounding the chromophore that is located in a central alpha helix. Some Green Fluorescent Proteins-like (GFP-like) proteins fold into the characteristic β-barrel shape but do not form a chromophore. The training set (pdb) is heavily weighted towards structures with a chromophore, less than 2% of the GFP-like structures in the pdb have no chromophore. There are 578 GFP-like proteins in the pdb, 568 of these structures have a fully formed chromophore, they are the GFP-like fluorescent proteins (GFP-FPs) and 10 have no chromophore.

The commonly accepted mechanism for chromophore formation in GFP-like proteins is shown in Fig 1. In order for the autocatalytic chromphore formation to occur the immature GFP-like protein has to be in the tight-turn conformation (Fig 1I) [22]. This tight turn breaks the canonical i to i+4 hydrogen bonding arrangement found in alpha helices [23], forming a kink that removes the intra main-chain hydrogen bonds that are commonly associated with an α-helix (S1 Fig). It is presumed that this aids in chromophore formation because the hydrogen bonds would otherwise have to be broken during maturation. Tyr66, Gly65, Arg96 and Glu222 (GFP numbering) are involved in chromophore formation and are highly conserved in all naturally occurring GFP-like FPs [24, 25]. The central tyrosine of the chromophore, Tyr66, is conserved in all naturally-occurring GFP-like FPs, although any aromatic residue in that position will auto-catalytically form a chromophore [24]. Thus, we have chemical knowledge that can be used to predict whether a GFP-like protein will form a chromophore or not. A GFP-like protein has to have a glycine65, an aromatic residue in position 66 and an arginine96 to form a chromophore. This is chemical knowledge that was not directly provided to AlphaFold2 or RoseTTAFold.

We have used AlphaFold2 and RoseTTAFold to generate the structures of two series of GFP-like sequences; one that can form a chromophore and the other that can't. And have shown that both programs fold the chromophore forming GFP-like sequences into a distinctly different conformation compared to the sequences that cannot form a chromophore. The structures in the training set taught the AI programs chemistry to distinguish which proteins will be post-translationally modified from those that will not. This was highly unexpected.

## Materials and methods

### Sequence selection

**Standard structures.**   Three GFP-like proteins with known solid-state structures were selected: 1EMA, 2AWJ, 1H4U (pdb codes).

1EMA is the crystal structure of the S65T mutant of GFP. Together with 1GFL [27] it was the first crystal structure of GFP to be solved [28]. It is the structure of a fluorescent protein that has a mature chromophore formed from the 65TYG67 triad of residues (Fig 1V).

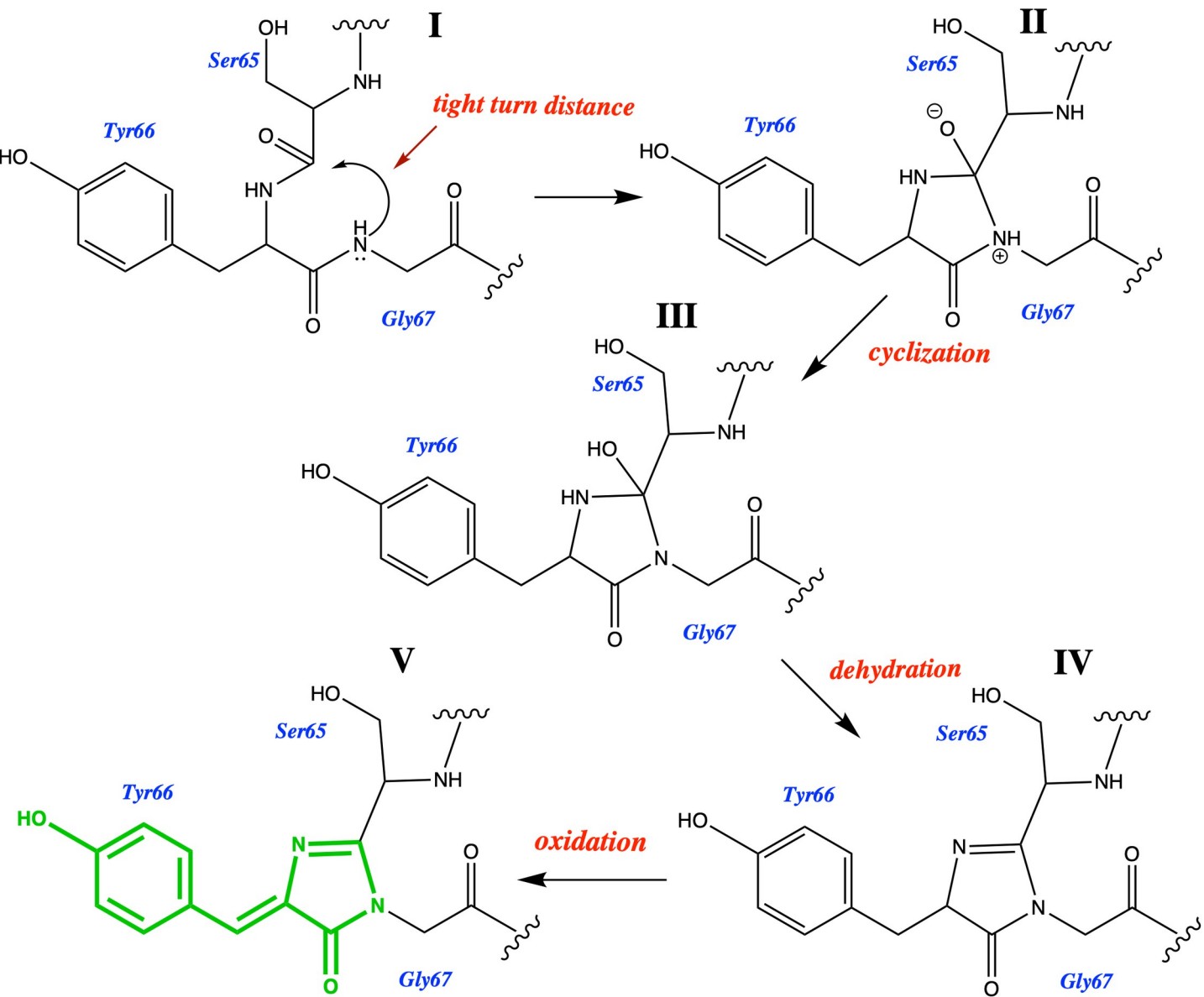

**Fig 1. The cyclization-dehydration-oxidation pathway for chromophore formation.** [26] Structure **I** is known as the immature precyclized form of the protein. The tight-turn distance depicted in structure **I** was measured for all sequences and is presented in S1 Data. Structure **V** shows the fully formed mature chromophore (green).

2AWJ is an immature precyclized GFP-like protein structure. It is the R96M mutant that takes about 3 months to form the chromophore [29]. Arginine and methionine have similar sizes, allowing us to safely assume that the conformation of immature wild-type GFP and its R96M mutant would be the same. The crystal structure of the R96M mutant is the closest solid-state structure to the immature GFP precyclized structure (Fig 1**I**).

1H4U is the structure of the G2 domain of mouse nidogen. It is a GFP-like structure with a central alpha helix in an 11 stranded beta barrel. It has no chromophore and cannot form a chromophore [30].

**Sequences of chromophore forming GFP-like proteins.** Amino acid sequences of GFP-like fluorescent proteins that do not appear in the pdb were obtained from FPbase [31]. The

chromophore forming GFP-like FP sequences were chosen from the FPBase in a way that they spanned as much phylogenetic space possible and had not been crystalized i.e. they did not appear in the pdb. Protein sequences were selected from each FP lineage, starting from the parent protein. If the parent protein's crystal structure was already solved, then the next protein sequence in the lineage closest to the parent was selected. No sequence was used from lineages in which all structures have been solved (DrCBD) and far-red FPs with a tetrapyrroloid chromphores as these FPs are not GFP-like β-barrels.

**Sequences of GFP-like proteins that cannot form a chromophore.** Sequences of GFP-like non-fluorescent proteins were taken from Haddock's "Non-excitable fluorescent protein orthologs found in ctenophores" paper [32] and a series of nidogen-G2 domains were obtained from a standard protein BLAST search (BLASTP) using the 1H4U FASTA sequence as a protein query (The KAB1270852.1 Nidogen-1 *Camelus dromedarius* sequence was chosen to honor the fact that the Connecticut College mascot is the camel). Only the G2 fragment region of the nidogen sequences were used in the AI alignments and the subsequent analyses. See S1 Table for sequences studied and their sequence similarity to S65T GFP (1EMA).

## AI predictions

RoseTTAFold was accessed via the RosettaCommons web-interface server Robetta [33] and simplified AlphaFold(v2.1.0) was accessed through AlphaFold's Colab notebook [34].

## Alignments

MAFFT DASH [35] was used for the multiple alignment and Clustal Omega [36] to find the percent identity matrix of all of the proteins relative to 1EMA, see S1 Table. All residues aligned with residues 60 to 74 of 1EMA were considered part of the central alpha-helix (S2 Fig).

## Measurements

Maestro Version 12.9.123 [37] was used to perform hydrogen bond measurements of the structures obtained from pdb [5], Alphafold2 and RoseTTAFold. RMSD values for the alpha helical overlaps were obtained by determining the pairwise distances between structures, using the rms displacement after optimal rigid-body superposition between pairs of non-hydrogen backbone atoms of residues highlighted in the alignment shown in S2 Fig of the supplementary material [38, 39]. All the measurements are presented in S1 Data.

## Results and discussion

### Structures examined

AlphaFold2 and RoseTTAFold were used to predict the 3-dimensional structure of a series of GFP-like proteins—a set of 3 standard structures with known solid state structures; a group of 21 GFP-like fluorescent proteins that will post-translationally form a chromophore and whose solid state structures have not been determined; as well as a group of 23 GFP-like non-fluorescent proteins that do not have the residues required to form a chromophore and whose solid state structures have not been determined. S1 Table lists all the sequences and how they were obtained.

### How does AI deal with the chromophore?

Since the chromophore formation involves post-translational modification of the 65TYG67 triad we can't expect the AI programs to model the structure of GFP-like FPs with a fully formed chromophore (Structure **V** in Fig 1). The central alpha-helical strand (residues 60 to

74) of the structure obtained from the AI predicted folding of the S65T GFP sequence (as found in 1EMA) was overlapped with the pdb structure of 1EMA. Both AlphaFold2 and RoseTTAFold predict a structure closest to the immature precyclized form (structure **I** in Fig 1, a structure similar to the one obtained if one was to graphically mutate the chromophore back to 65TYG67 and then minimize), a conformation most closely represented by crystal structure of the R96A mutant 2AWJ (RMSD for AF2 = 0.32Å and RF = 0.40Å). The alpha helical overlaps with the regular alpha helix observed in the nidogen G2 domains (RMSD vs 1H4U for AF2 = 1.65Å and RF = 1.68Å) or even vs. its own crystal structure (RMSD vs 1EMA for AF2 = 0.78Å and RF = 0.80Å) are much higher, see S1 Data.

## Can AI folding programs predict whether GFP-like proteins will form a chromophore or not?

AlphaFold2 and RoseTTAFold were used to predict the 3-dimensional structures of 21 GFP-like fluorescent proteins that will post-translationally form a chromophore as well as a group of 23 GFP-like non-fluorescent proteins that do not have the residues required to form a chromophore. All sequences folded into GFP-like 11 stranded β-barrels with a central alpha helix. The resultant structures were analyzed to establish whether according to the chromophore forming mechanism discussed in the introduction and shown in Fig 1, they were in conformations geometrically primed for chromophore formation or whether they could not form a chromophore. We overlapped their central alpha helical strands with the strands obtained from the crystal structures of 1EMA (has chromophore), 2AWJ (has no chromophore but is in the right conformation to form one) and 1H4U (has no chromophore and is in the wrong conformation to form chromophore); their tight-turns were measured, as were the i to i+4 hydrogen bonding distances in their alpha helices.

**Statistical analysis of the AlphaFold2 predictions.** From the boxplot, Fig 2, as well as the data presented in S1 Data and S2 Table, it is apparent that the RMSD for α-helix overlap of the AlphaFold2 predicted structures with 1EMA-crystal is much higher for those that will not form a chromophore than those that do. Moreover, the Welch two sample t-test shows there is a significant difference in the mean values of α-helix overlap with 1EMA-crystal between GFP-like

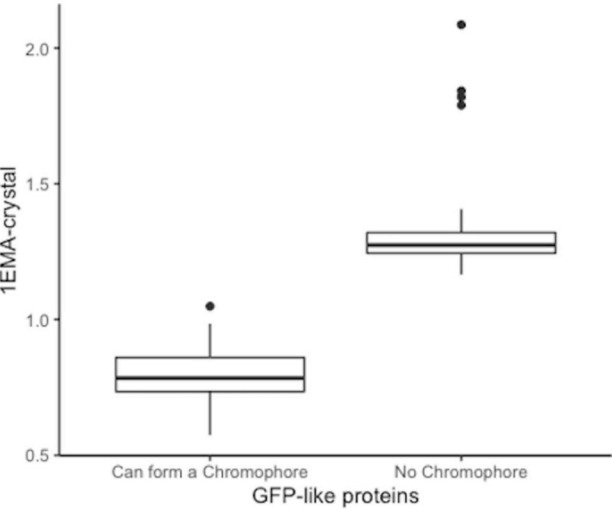

**Fig 2. RMSD overlap of the α-helix of the 1EMA-crystal structure with the α-helix of 1EMA as determined by AlphaFold2 for GFP-like proteins that will form a chromophore (SYG) and those that do not (no SYG).**

proteins that will form a chromophore and those that do not (t-value = 10.646 and p-value = 7.77 x $10^{-12}$). The average RMSD for GFP-like proteins that will not form a chromophore is 1.372Å while the average value for GFP-like proteins that will form a chromophore is 0.799Å.

For the structures predicted by AlphaFold2 one only needs to compare the predicted alpha helical structure with that of the S65T GFP pdb structure to know whether the GFP-like protein will form a chromophore or not.

A LASSO regression model was built using "can form chromophore" (= 1) and "cannot form chromophore" (= 0) as the response variable with all the measurements including Alpha helix overlap, tight turn distance, as well as H-Bond distances in angstrom. The final model includes the following predictors—the alpha helix overlap with 1EMA-crystal as well as H-bond distances in Angstrom between residues 61–65 (HD2), 62–66 (HD3), 70–74 (HD11). The model shows there is a clear distinction between the sequences capable of chromophore forming and those that don't have the residues required for chromophore formation, see S3 Fig and S3 Table. A model built on just the measurements of the hydrogen bonding distances is also able to distinguish between chromophore and non-chromophore forming structures.

**Statistical analysis of the RoseTTAFold predictions.** The RMSD for the α-helix overlap of the RoseTTAFold predicted structures with 1EMA-crystal structures is generally higher for those structures that will not form a chromophore than those that do. However the appropriate boxplot (S4 Fig) as well as the data presented in S1 Data and S4 Table, show that while RoseT-TAFold can distinguish between posttranslationally modified and non-posttranslationally structures it is not as distinctive as AlphaFold2.

The Welch two sample t-test shows that there is a significant difference in the mean values for alpha helix overlap with 1EMA-crystal between GFP-like proteins that will form a chromophore and those that do not (t-value = 4.699 and p-value = 3.526 x $10^{-5}$). The average value for GFP-like proteins that will not form a chromophore is 1.317 while the average value for GFP-like proteins that will form a chromophore is 1.003.

For data collected using RoseTTaFold, we built a LASSO regression model using "can form chromophore" (= 1) and "cannot form chromophore" (= 0) as the response variable. The selected predictors of final model are: the RMSD of the alpha helix overlap with 1EMA-crystal and 2AWJ-crystal, the tight turn distances, as well as H-bond distances in Angstrom between residues 61 and 65 (HD2), 62–66 (HD3), 63–67 (HD4), 64–68 (HD5), see S5 Table and S5 Fig. From the prediction results, we can see there is a clear distinction between the sequences capable of forming a chromophore and those that can't. A model built on just the measurements of hydrogen bonding distances is also able to distinguish between chromophore and non-chromophore forming structures.

## Conclusions

Only 10 of the 578 GFP-like proteins in the pdb have no chromophore, yet when AlphaFold2 and RoseTTAFold are presented with the sequences of 44 GFP-like proteins that are not in the pdb they fold the proteins in such a way that one can unequivocally distinguish between those that can and cannot form a chromophore. They predict the conformation of the immature protein with a kink in the α-helix (as expected from machine learning vs. memorization) and have used their training set to learn some chemistry, they can distinguish between GFP-like proteins that will form a chromophore and those that do not. We suspect that the pdb training set and multiple sequence alignments enable AlphaFold2 and RoseTTAFold to "think" like chemists and look for the presence of residues equivalent to Arg96, Gly65 and an aromatic residue at position 66 in GFP–the residues required for chromophore formation–and use those to fold the sequence in a chromophore forming or non-chromophore forming conformations.

## Supporting information

**S1 Fig. Main chain i to i+4 "hydrogen bonding" alpha helical interactions measured and presented in S1 Data.**
(DOCX)

**S2 Fig. Alignment of all sequences used.**
(DOCX)

**S3 Fig. Predicted possibility that structure forms a chromophore or cannot form a chromophore using LASSO model from S2 Table on AlphaFold2 data.**
(DOCX)

**S4 Fig. RMSD overlap of the α-helix of the 1EMA-crystal with the α-helix of 1EMA as determined by RoseTTAFold for GFP-like proteins that will form a chromophore and those that do not.**
(DOCX)

**S5 Fig. Predicted possibility that structure forms a chromophore or cannot form a chromophore using LASSO model from S4 Table, which is based on RoseTTAFold data.**
(DOCX)

**S1 Data. RMSD overlap of the α-helix of the 1EMA, 2AWJ, 1H4U crystal structures with the α-helix of the RoseTTAFold and AlphaFold determined structures, tight turn distances of RoseTTAFold and AlphaFold determined structures and the main chain i to i+4 "hydrogen bonding" alpha helical interactions.** These are the geometric measurements that are crucial to chromophore formation. The structures are divided into two groups, GFP-like proteins that will form a chromophore and those that do not.
(XLSX)

**S1 Table. Description of all sequences used in this study [28–30, 32, 40–62].**
(DOCX)

**S2 Table. Summary statistics of the RMSD overlap (in Angstrom) of the α-helix of the 1EMA-crystal with the α-helix of 1EMA as determined by AlphaFold2 for GFP-like proteins that will form a chromophore and those that do not.**
(DOCX)

**S3 Table. LASSO model results for the alpha helix overlap with 1EMA-crystal as well as H-bond distances in angstrom between residues 61–65 (HD2), 62–66 (HD3), 70–74 (HD11) collected using AlphaFold2.**
(DOCX)

**S4 Table. Summary statistics of the RMSD overlap (in Angstrom) of the α-helix of the 1EMA-crystal with the α-helix of 1EMA as determined by RoseTTAFold for GFP-like proteins that will form a chromophore and those that do not.**
(DOCX)

**S5 Table. LASSO model results for the RMSD of the alpha helix overlap with 1EMA and 2AWJ-crystal structures, the tight turn distances, as well as H-bond distances in Angstrom between residues 61 and 65 (HD2), 62–66 (HD3), 63–67 (HD4), 64–68 (HD5) collected using RoseTTAFold simulations.**
(DOCX)

## Author Contributions

**Conceptualization:** Marc Zimmer.

**Data curation:** Sophia M. Hartley, Kelly A. Tiernan, Gjina Ahmetaj, Adriana Cretu.

**Formal analysis:** Sophia M. Hartley, Kelly A. Tiernan, Gjina Ahmetaj, Adriana Cretu, Yan Zhuang.

**Investigation:** Sophia M. Hartley, Kelly A. Tiernan, Gjina Ahmetaj, Adriana Cretu, Marc Zimmer.

**Methodology:** Marc Zimmer.

**Project administration:** Marc Zimmer.

**Resources:** Marc Zimmer.

**Software:** Marc Zimmer.

**Supervision:** Marc Zimmer.

**Validation:** Yan Zhuang.

**Writing – original draft:** Marc Zimmer.

**Writing – review & editing:** Marc Zimmer.

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
