## [Decision Letter · Decision Letter 0]

31 Mar 2022

PONE-D-22-04782AlphaFold2 and RoseTTAFold Predict Posttranslational modifications. Chromophore Formation in GFP-like Proteins.PLOS ONE

Dear Dr. Marc Zimmer,

Thank you for submitting your manuscript to PLOS ONE. Apologies for the delay in getting back to you. Seven reviewers were contacted and out of them only one answered. After careful consideration of the review, we feel that your manuscript has merit but needs the minor amendements denoted by the reviewer to fully meet PLOS ONE’s publication criteria . Therefore, we invite you to submit a revised version of the manuscript that convincingly addresses the points raised during the review process.

We look forward to receiving your revised manuscript.

Kind regards,

Maria Gasset, Ph.D.

Academic Editor

PLOS ONE

Journal Requirements:

2. PLOS requires an ORCID iD for the corresponding author in Editorial Manager on papers submitted after December 6th, 2016. Please ensure that you have an ORCID iD and that it is validated in Editorial Manager. To do this, go to ‘Update my Information’ (in the upper left-hand corner of the main menu), and click on the Fetch/Validate link next to the ORCID field. This will take you to the ORCID site and allow you to create a new iD or authenticate a pre-existing iD in Editorial Manager. Please see the following video for instructions on linking an ORCID iD to your Editorial Manager account: https://www.youtube.com/watch?v=_xcclfuvtxQ.

Reviewers' comments:

Reviewer's Responses to Questions

**Comments to the Author**

1. Is the manuscript technically sound, and do the data support the conclusions?

Reviewer #1: Yes

2. Has the statistical analysis been performed appropriately and rigorously? 

Reviewer #1: I Don't Know

3. Have the authors made all data underlying the findings in their manuscript fully available?

Reviewer #1: Yes

4. Is the manuscript presented in an intelligible fashion and written in standard English?

Reviewer #1: Yes

5. Review Comments to the Author

Reviewer #1: The authors report on a very interesting computational approach to figure out whether green fluorescent protein-like proteins fluoresce, based on their amino acid sequence. They use the AlphaFold2 and RoseTTAFold artificial intelligence software which was trained on 180,000 protein structures in the protein data bank to predict the folded protein with atomic resolution. This data base contains 578 GFP-like proteins of which 10 have no chromophore, i.e. do not fluoresce. (It wasn’t clear to me whether they can still absorb, though, like a dark acceptor in FRET?)

The present manuscript presents the amino acid sequences of 44 GFP-like proteins that are not in the protein data bank to AlphaFold2 and RoseTTAFold, and the software calculates the 3-dimensional structure of these proteins to atomic resolution. While the precise solid-state structure of these 44 proteins has not been determined experimentally, importantly, it is known from experiment whether they fluoresce or not.

The software correctly folds the amino acid sequences into a fluorophore for the fluorescent proteins, and no fluorophore for the ones that do not fluoresce, if I understand this correctly.

I think is a phenomenal result, and the link to experiment via fluorescence is ingenious – it requires no details apart from “yes” or “no” of fluorescence, and no details of spectra or lifetime, which essentially makes it a powerful binary approach (0 or 1, so to speak).

The manuscript is very well written and easy to follow. The work will be of interest to the research community developing (and also using) fluorescent proteins, and those predicting protein structure from amino acid sequences in general. It will also be of interest for researchers in artificial intelligence, machine learning and other computational and computer-simulation approaches.

I only have some minor comments:

1) Maybe a figure with a representative structure of a GFP-like protein with fluorophore investigated in this study would be useful, as well as such a protein without a fluorophore.

2) By chromophore, the authors mean a structure that fluoresces, not one that just absorbs light, is that correct?

6. PLOS authors have the option to publish the peer review history of their article (what does this mean?). If published, this will include your full peer review and any attached files.

Reviewer #1: No

---

## [Author Response · Author response to Decision Letter 0]

6 Apr 2022

We were pleased to see that the reviewer was so positive about the manuscript and only had two minor suggestions. Here are our responses to the suggestions.

1) Maybe a figure with a representative structure of a GFP-like protein with fluorophore investigated in this study would be useful, as well as such a protein without a fluorophore.

I don’t think a figure such as the one suggested above will contribute much to the understanding of the work presented. They are essentially the same because the deep learning programs show the conformation prior to post-translational cyclization and do not show the mature chromophore. However, we would be happy to insert a figure if desired. (See response to reviewers .doc file)

2) By chromophore, the authors mean a structure that fluoresces, not one that just absorbs light, is that correct?

Yes, we have clarified this by adding the following definition at the first occurrence of the word chromophore. “chromophore (the part of the protein responsible for fluorescence)”.

---

## [Decision Letter · Decision Letter 1]

12 Apr 2022

AlphaFold2 and RoseTTAFold Predict Posttranslational modifications. Chromophore Formation in GFP-like Proteins.

PONE-D-22-04782R1

Dear Dr. Marc Zimmer,

We’re pleased to inform you that your manuscript has been judged scientifically suitable for publication and will be formally accepted for publication once it meets all outstanding technical requirements.

Kind regards,

Maria Gasset, Ph.D.

Academic Editor

PLOS ONE

Additional Editor Comments (optional):

Reviewers' comments:

Reviewer's Responses to Questions

**Comments to the Author**

1. If the authors have adequately addressed your comments raised in a previous round of review and you feel that this manuscript is now acceptable for publication, you may indicate that here to bypass the “Comments to the Author” section, enter your conflict of interest statement in the “Confidential to Editor” section, and submit your "Accept" recommendation.

Reviewer #1: All comments have been addressed

2. Is the manuscript technically sound, and do the data support the conclusions?

Reviewer #1: Yes

3. Has the statistical analysis been performed appropriately and rigorously? 

Reviewer #1: I Don't Know

4. Have the authors made all data underlying the findings in their manuscript fully available?

Reviewer #1: Yes

5. Is the manuscript presented in an intelligible fashion and written in standard English?

Reviewer #1: Yes

6. Review Comments to the Author

Reviewer #1: Now I agree that a figure of a fluorescent and non-fluorescent GFP would be of limited use, as it is dominated by the barrel, not the all-important fluorophore. I leave it to the authors to decide whether to include this figure or not, it may be helpful for non-experts in the field. The chromophore definition is definitely useful, and I look forward to seeing this nice work published.

7. PLOS authors have the option to publish the peer review history of their article (what does this mean?). If published, this will include your full peer review and any attached files.

Reviewer #1: No

---

## [Editor Report · Acceptance letter]

27 May 2022

PONE-D-22-04782R1 

AlphaFold2 and RoseTTAFold predict posttranslational modifications. Chromophore formation in GFP-like proteins. 

Dear Dr. Zimmer:

I'm pleased to inform you that your manuscript has been deemed suitable for publication in PLOS ONE. Congratulations! Your manuscript is now with our production department. 

Kind regards, 

on behalf of

Dr. Maria Gasset 

Academic Editor

PLOS ONE